# The Role of Imaging in Monitoring Large Vessel Vasculitis: A Comprehensive Review

**DOI:** 10.3390/biom15111505

**Published:** 2025-10-24

**Authors:** Inês Sopa, Roberto Pereira da Costa, Joana Martins Martinho, Cristina Ponte

**Affiliations:** 1Rheumatology Department, Unidade Local de Saúde Santa Maria, Centro Académico de Medicina de Lisboa (CAML), 1649-028 Lisbon, Portugal; robertopc25@hotmail.com (R.P.d.C.); joanamartinsmartinho@gmail.com (J.M.M.); cristinadbponte@gmail.com (C.P.); 2Faculdade de Medicina, Universidade de Lisboa, Centro Académico de Medicina de Lisboa (CAML), 1649-028 Lisbon, Portugal

**Keywords:** large vessel vasculitis (LVV), giant cell arteritis (GCA), Takayasu arteritis (TAK), ultrasound, MRI (magnetic resonance imaging), PET (positron emission tomography), CT (computed tomography), disease monitoring, disease activity, biomarkers

## Abstract

Giant cell arteritis (GCA) and Takayasu arteritis (TAK) are forms of primary large vessel vasculitis (LVV) affecting the aorta and its major branches. Timely diagnosis and accurate monitoring are essential to prevent irreversible damage. Current assessment strategies rely heavily on symptoms, physical examination, and inflammatory markers, which lack sensitivity and specificity, particularly in patients treated with IL-6 inhibitors. This narrative review provides a comprehensive overview of the role of imaging in monitoring LVV. Ultrasound, magnetic resonance imaging, and positron emission tomography better reflect disease activity and treatment response compared to conventional clinical and laboratory measures. Notably, emerging imaging-based tools such as the OMERACT GCA Ultrasound Score, the Takayasu Ultrasound Index, and the TAK Integrated Disease Activity Index (TAIDAI) are promising treat-to-target instruments. While computed tomography is primarily used to assess structural damage, conventional angiography now plays a more limited role, mainly reserved for procedural planning and haemodynamic evaluation. A key challenge remains: interpreting persistent vascular abnormalities, which may indicate active disease, vascular remodelling, or irreversible damage. Standardisation of imaging protocols and interpretation is needed, alongside further research on the prognostic value of imaging for relapse risk. This review supports a multimodal, patient-tailored approach in which imaging is central to the long-term management of LVV.

## 1. Introduction

Large vessel vasculitis (LVV), encompassing primarily giant cell arteritis (GCA) and Takayasu arteritis (TAK), is a group of rare inflammatory diseases affecting the aorta and its major branches. Accurate diagnosis and effective monitoring are critical to prevent irreversible complications such as stenosis, aneurysms, or ischaemic events including vision loss and stroke. Traditionally, clinicians have relied on clinical assessment and laboratory markers like erythrocyte sedimentation rate (ESR) and C-reactive protein (CRP), which lack specificity and sensitivity for vascular inflammation, especially in patients under immunosuppressive treatment with IL-6 inhibitors.

Imaging has emerged as a key tool in both diagnosis and disease monitoring. Modalities such as ultrasound, magnetic resonance imaging (MRI), positron emission tomography (PET), computed tomography (CT), and conventional angiography offer complementary advantages in assessing vascular involvement. However, their roles in follow-up and relapse prediction continue to evolve, and integrating these tools into standard monitoring strategies remains an area of active investigation.

This narrative review provides a comprehensive overview of the current evidence on the role of imaging in monitoring large vessel vasculitis. We explore the applications, strengths, and limitations of each modality in both GCA and TAK, highlighting their utility in assessing disease activity, structural damage, and therapeutic response. We also discuss emerging technologies and potential directions for standardisation, aiming to support clinicians and researchers in integrating imaging more effectively into the longitudinal management of LVV. The scope of this review is limited to adult-onset GCA and TAK and does not extend to paediatric or atypical presentations.

## 2. Ultrasound

Ultrasound is a valuable imaging modality for assessing LVV, offering a non-invasive, radiation-free, and cost-effective tool for real-time vascular evaluation. It allows for the identification of characteristic inflammatory changes, most notably the halo sign—a well-defined, hypoechoic, and concentric thickening of the arterial wall adjacent to the lumen, reflecting vessel wall oedema and active inflammation in GCA [1,2] (see Figure 1). In Takayasu arteritis (TAK), the macaroni sign—a circumferential, homogeneous, mid-echoic thickening of the arterial wall, particularly in the common carotid arteries—is frequently observed and represents a pathophysiological process analogous to the halo sign in GCA [3] (see Figure 2). Although these sonographic signs are highly characteristic of vascular inflammation, false positives can occur, most often due to overlap with atherosclerotic changes [4]. This underscores the importance of operator expertise and standardised protocols, while also highlighting the increased cardiovascular burden in LVV patients, in whom accelerated atheromatosis and arteriosclerosis contribute to excess morbidity and mortality [5]. Beyond the halo and macaroni signs, ultrasound can also detect complications such as stenoses, occlusions, and aneurysms, which are relevant in both GCA and TAK. While particularly effective for imaging superficial arteries, its utility in evaluating deeper vessels, such as the thoracic aorta, is limited, and the diagnostic yield remains highly operator-dependent.

### 2.1. Giant Cell Arteritis

In suspected GCA, ultrasound of the temporal (TA) and axillary (AX) arteries is currently regarded as the first-line imaging modality [1]. Once the diagnosis is established, monitoring disease activity becomes essential, with ultrasound playing a central role in assessing vascular inflammation and treatment response. Recent studies have explored the persistence of the halo sign and longitudinal changes in intima-media thickness (IMT) as imaging biomarkers for follow-up [6,7,8,9,10,11,12].

Prospective data from Ponte et al. demonstrated that TA halo characteristics—particularly IMT and the number of involved segments—respond to glucocorticoid therapy and correlate with inflammatory markers such as CRP and ESR, whereas AX changes tend to persist longer, likely reflecting structural remodelling. Notably, the halo sign was still detected in 94% of first relapses, albeit with a lower burden than at diagnosis [13]. In a separate prospective cohort, Haaversen et al. reported moderate diagnostic performance of ultrasound for detecting relapses (sensitivity ~61%, specificity ~72%) [11]. Similarly, Bosch et al. observed a gradual reduction in AX IMT over time and proposed IMT thresholds to distinguish chronic from active axillary vasculitis in a mixed retrospective and prospective GCA cohort [14].

Ultrasound remains informative in patients treated with IL-6 inhibitors, in whom suppression of systemic inflammatory markers limits the reliability of laboratory monitoring. Seitz et al. and Sebastian et al. showed that findings such as halo persistence and changes in IMT remain responsive despite low CRP and ESR levels, thereby providing an objective measure of disease activity [6,8]. This makes ultrasound particularly valuable for relapse detection when laboratory results are inconclusive. It should be noted, however, that not all relapses are accompanied by vascular changes—those manifesting with isolated polymyalgia rheumatica (PMR) symptoms, for instance, may occur without detectable sonographic abnormalities. Such relapses are generally considered less severe and are unlikely to reflect active vascular inflammation [13].

To further standardise vascular assessment, ultrasound-based scoring systems such as the OMERACT GCA Ultrasound Score (OGUS) have been developed. OGUS aggregates IMT measurements across eight arterial segments and has demonstrated strong inter- and intra-observer reliability, high sensitivity to change, and moderate correlation with inflammatory biomarkers [15]. Molina-Collada et al. and Schäfer et al. showed that OGUS improves in patients achieving remission, whereas a stable or increasing score was associated with relapse [12,16]. In a 15-month follow-up study, Nielsen et al. confirmed that OGUS and composite TA scores could differentiate remission from active disease, with several metrics achieving AUC values above 0.8 [10]. Most notably, a multicentre prospective study by Monti et al. demonstrated the prognostic value of OGUS: higher baseline scores predicted increased relapse risk at 12 months, while early improvement or normalisation of OGUS within the first 3–6 weeks was independently associated with a significantly lower likelihood of relapse [17].

Together, these findings position OGUS as a promising treat-to-target instrument in GCA—capable of guiding therapeutic decisions, stratifying relapse risk, and providing reliable measures of disease activity, even in the context of IL-6 blockade. Nonetheless, despite the growing body of evidence, current EULAR recommendations continue to limit the use of ultrasound in routine follow-up, advocating its application primarily in suspected relapses rather than for systematic monitoring in patients in clinical and biochemical remission [18]. This reflects ongoing concerns regarding inter-operator variability and the risk of overinterpreting residual vascular abnormalities. Establishing individualised “ultrasound remission baselines” [19] may enhance longitudinal interpretation; however, the true added value of routine ultrasound monitoring over traditional clinical and laboratory approaches remains to be fully validated.

### 2.2. Takayasu Arteritis

While MRI remains the imaging modality of choice for assessing the thoracic aorta and its branches in TAK, ultrasound—particularly contrast-enhanced ultrasound (CEUS)—is increasingly utilised to evaluate more accessible arteries, such as the common carotid and subclavian arteries, which are among the most frequently involved vascular territories in TAK [20,21]. A meta-analysis by Barra et al. reported a pooled sensitivity of 81% and specificity above 90% for ultrasound in diagnosing TAK, although most studies were small and focused on carotid involvement [22]. Beyond diagnosis, ultrasound is gaining relevance in monitoring disease activity [23,24,25]. Increased intima-media thickness (IMT), particularly in the carotid arteries, has been associated with active inflammation, although standardised cut-offs remain undefined. A threshold of 1 mm is often used but may be confounded by age and atherosclerosis [26,27]. To address this, Svensson et al. proposed the Takayasu Ultrasound Index, which averages maximum IMT across large arteries such as the common carotids, brachiocephalic trunk, and aortic arch, showing correlation with clinical activity and potential for longitudinal monitoring [28].

CEUS allows for the detection of arterial wall neovascularisation, serving as an indirect marker of inflammation. Early case reports by Giordana et al., Magnoni et al., and Possemato et al. illustrated CEUS’s capacity to detect inflammatory activity in the carotid arteries, with enhancement changes reflecting treatment response [29,30,31]. Herlin et al. also described persistent CEUS enhancement in a patient with normalised inflammatory markers, highlighting its sensitivity to subclinical inflammation [32]. In a small prospective case series, Schinkel et al. demonstrated that CEUS allows for a dynamic assessment of carotid wall vascularisation in patients with TAK and GCA [33]. In an observational study, Germanò et al. reported strong concordance between CEUS vascularisation grades, PET-CT uptake, and disease activity as defined by the Kerr criteria, supporting its utility in identifying active disease [34]. In a separate prospective study, Ma et al. showed that CEUS could detect persistent neovascularisation even in patients considered clinically inactive, suggesting that inflammation may persist despite biochemical remission [26]. Wang et al., also in a prospective study, found that CEUS enhancement grades correlated with ESR, IMT, and disease activity, achieving an AUC of 0.87 for detecting active disease [35]. Most recently, Dong et al., in a large prospective cohort, demonstrated that CEUS enhancement and wall thickness declined with treatment and more accurately reflected disease activity than acute-phase reactants, even in patients with widespread arterial involvement (types II and V) [27].

A novel approach using ultrasound localisation microscopy (ULM) was introduced by Goudot et al., enabling visualisation of microvascular flow in the arterial wall at a micron scale. In a pilot study with three active and eleven quiescent TAK patients, microbubble density in the carotid wall was significantly higher in active compared to inactive disease, suggesting its potential as a precise, non-invasive marker of activity [36].

In summary, ultrasound, particularly CEUS, is a promising tool for monitoring disease activity in TAK, offering a non-invasive, repeatable method for assessing inflammation in key vascular territories. While not a substitute for MRI or PET in comprehensive disease assessment, it provides complementary information for evaluating treatment response. Further prospective studies and standardised scoring systems are needed to support its integration into routine longitudinal care.

## 3. Magnetic Resonance Imaging

MRI, including magnetic resonance angiography (MRA), provides a comprehensive, radiation-free method for evaluating vascular inflammation in large vessel vasculitis (LVV). It allows for direct visualisation of both the vessel wall and the lumen, enabling detection of mural thickening, oedema, contrast enhancement, and luminal changes such as stenosis or aneurysms. T1-weighted sequences highlight vessel wall morphology and contrast enhancement, while T2-weighted or STIR sequences can identify perivascular oedema—an indirect marker of active inflammation. Gadolinium-based contrast agents further enhance sensitivity for detecting vascular wall pathology [37,38,39]. Unlike 18F-FDG PET, MRI avoids ionising radiation and is preferred for younger patients and in settings where repeated imaging is necessary. However, its utility is constrained by limited accessibility, high cost, and the need for specialised radiological expertise. Additional limitations include artefacts from motion or calcification, contraindications in patients with non-MRI-compatible implants or devices, and challenges in differentiating inflammation from chronic damage, particularly in the context of vessel remodelling [18,40].

### 3.1. Giant Cell Arteritis

MRI has demonstrated diagnostic value in both extracranial GCA [41,42,43,44,45] and cranial GCA [46,47,48]. Recent efforts to standardise interpretation include the MRVAS score proposed by Froehlich et al. and orbital vessel wall imaging protocols introduced by Rhee et al. [49,50]. Building on these diagnostic advances, several studies have explored the utility of MRI for monitoring disease activity over time, particularly in patients undergoing immunosuppressive treatment or considered to be in clinical remission.

In an early observational study, Both et al. reported discordance between PET and MRI findings in treated patients with large vessel GCA (LV-GCA), suggesting that MRI may detect residual vessel wall inflammation not captured by metabolic imaging or acute-phase reactants. This raised early interest in MRI as a tool for uncovering subclinical disease [43]. In a retrospective cohort, Adler et al. observed that mural enhancement on MRI typically decreases following immunosuppressive therapy but may persist in some cases. In a subsequent prospective cohort, they evaluated patients following tocilizumab discontinuation and found that residual enhancement did not predict relapse, highlighting the need for caution when interpreting persistent imaging abnormalities [44]. Quinn et al., in a prospective head-to-head study, compared MRI and PET in patients in clinical remission and found that approximately half exhibited signs of vascular activity on imaging, despite normal inflammatory markers [51]. These findings further supported MRI’s ability to identify possible subclinical inflammation.

The importance of serial MRI was demonstrated by Reichenbach et al., in a prospective study nested within a randomised controlled trial. Patients with positive baseline MRA underwent repeat imaging at weeks 12 and 52, revealing that wall thickening and enhancement frequently persisted despite clinical remission [42]. Although not necessarily markers of active disease, these findings may reflect structural progression and support the role of MRI in detecting new stenoses or aneurysms. Similarly, Christ et al., in a prospective cohort with scheduled MRI assessments, followed patients receiving ultra-short glucocorticoids and tocilizumab. They reported resolution of cranial vessel enhancement after 52 weeks in most cases, while mural enhancement in large vessels and PMR-related regions often persisted despite clinical and biochemical remission [52]. These findings suggest that cranial MRI may be more responsive to change and potentially useful in monitoring treatment response and guiding long-term follow-up, though further validation is warranted.

### 3.2. Takayasu Arteritis

In TAK, MRI—including contrast-enhanced MRA—is considered a first-line imaging modality, especially in younger patients requiring serial assessments. Its ability to detect vessel wall inflammation across the aorta and its major branches makes it particularly valuable in early disease (see Figure 3). While several studies [53,54,55] have confirmed its diagnostic utility, the role of MRI in disease monitoring remains more controversial.

Early observations from Eshet et al. revealed limited concordance between MRI findings and clinical relapses during long-term follow-up, raising questions about its sensitivity for capturing disease activity [56]. More recently, a prospective study by Sun et al. suggested that serial MRI may provide meaningful information in selected contexts. Sun et al. demonstrated that while luminal stenoses and wall thickness remained stable, vessel wall enhancement significantly decreased after immunosuppressive treatment—changes not mirrored by CRP or ESR levels, highlighting the ability of MRI to detect subclinical response [57]. In a cross-sectional study, Jiang et al. proposed a composite MRI score incorporating wall thickness, enhancement, and luminal narrowing, which showed moderate correlation with laboratory markers and clinical status [58]. Although not longitudinal, this framework contributed to the development of structured assessment tools. In this context, Tombetti et al. introduced an angiographic scoring system applicable to both MRA and CTA, validated primarily in TAK patients, and designed to enable consistent cross-sectional assessment of structural vascular damage across imaging modalities [59].

Despite these advances, significant challenges persist. As enhancement can reflect both inflammation and fibrosis or extracellular expansion, interpretation remains complex, particularly in the presence of longstanding vascular damage. Additionally, it remains unclear whether progression of luminal lesions, such as new stenoses, necessarily reflects active vasculitis or represents irreversible structural remodelling. These uncertainties are reflected in recent EULAR recommendations, which support MRI for diagnosis and evaluation of suspected relapses, but not for routine monitoring in patients in clinical remission [18,21]. Nonetheless, MRI remains central to the diagnostic workup of TAK, and serial imaging may be informative in selected cases, particularly when interpreted alongside standardised scoring systems. Further prospective validation is needed to clarify its role in guiding long-term disease management.

## 4. Fluorine-18-Fluorodeoxyglucose PET (FDG-PET)

FDG-PET is a functional imaging modality that reflects glucose uptake by pro-inflammatory cells with high glycolytic activity [60]. Its value was initially established in oncology and has since been extended to the evaluation of inflammatory and infectious diseases [60,61]. The major advantages of FDG-PET include the ability to assess multiple vascular territories in a single scan and to identify alternative diagnoses. However, disadvantages include relatively low spatial resolution, limited sensitivity for vascular complications, radiation exposure, high cost, and limited availability.

FDG-PET findings can be interpreted qualitatively, commonly through visual comparison of vascular uptake to reference tissues such as the liver, blood pool, or surrounding tissue, and expressed using scores such as the PET vascular activity score (PETVAS) or the total vascular score (TVS) [62,63]. These qualitative scores are widely used in clinical practice due to their ease of application and reproducibility. However, their objectivity remains debated, particularly in the context of longitudinal assessments and therapeutic monitoring. Alternatively, semiquantitative metrics, including standardised uptake values (SUVs) and target-to-background ratios (TBRs), may offer greater discriminatory power for evaluating disease activity and treatment response [62,64,65,66].

### 4.1. Giant Cell Arteritis

Traditionally, FDG-PET has been employed primarily for detecting extracranial involvement in GCA (see Figure 4). However, with advances in scanner resolution and image reconstruction techniques, its diagnostic accuracy for cranial GCA has improved considerably, with high specificity reported in recent studies and FDG-PET now incorporated into the 2023 EULAR recommendations for cranial assessment [18,67,68].

FDG-PET has increasingly been used to monitor disease activity in GCA, particularly in the context of immunosuppressive treatment. Multiple studies have demonstrated that FDG uptake typically decreases with therapy yet may persist in a proportion of patients [63,69,70]. Blockmans et al., in a prospective study, reported a significant reduction in TVS at three months after glucocorticoid initiation, with stabilisation at six months. Interestingly, the TVS at relapse was lower than at diagnosis [59]. Galli et al. found that PETVAS correlated with clinical disease activity, showing moderate accuracy in differentiating active disease [63]. Several other studies have confirmed reductions in the PETVAS, TVS, and TBR following treatment with glucocorticoids, methotrexate, or tocilizumab, with slight increases occasionally observed upon treatment tapering or discontinuation (e.g., tocilizumab) [66,71,72,73,74,75,76,77,78,79]. Schönau et al. observed a non-statistically significant higher reduction in PETVAS in patients treated with methotrexate or tocilizumab compared to those receiving prednisolone monotherapy [76]. Interestingly, Martínez-Rodríguez found that in cases of LVV where treatment was initiated without clinical improvement, the TBR did not show a significant decrease [74]. Although promising, the interpretation of persistent vascular FDG uptake remains challenging. Multiple studies report vascular uptake in patients with LVV, particularly GCA, who are clinically in remission [62,70,71,73,75,78]. Notably, up to 60% of such patients demonstrate uptake suggestive of active vasculitis [70,75]. The clinical significance of this persistent uptake remains uncertain, potentially reflecting subclinical inflammation, vascular remodelling, or atherosclerosis [63,70]. Consequently, current guidance suggests restricting FDG-PET to patients with suspected active disease, while its role in routine monitoring during remission remains unclear [18].

The prognostic relevance of FDG-PET is still debated [62,63,70,71]. Some studies, including those by Blockmans et al. and Sammel et al., reported no clear association between baseline FDG uptake and relapse risk [62,71]. In contrast, Grayson et al. found that a PETVAS ≥ 20 during clinical remission predicted relapse within 15 months in 56 patients with LVV, including 30 with GCA [70]. Froehlich et al. reported that higher baseline TBR values were associated with a lower risk of relapse in 21 patients with GCA, potentially indicating greater treatment responsiveness in individuals with initially higher inflammatory burden [80]. Similarly, Dellavedova et al. found that semiquantitative baseline FDG uptake in 17 patients with LVV correlated with persistent disease activity, vascular complications, and challenges in tapering glucocorticoids [81]. Moreover, FDG uptake has been implicated in the development of structural vascular damage. Studies by Moreel et al. and de Boysson et al. identified an association between FDG signal intensity and the subsequent formation of aortic dilatations or aneurysms [82,83].

Proper standardisation of scan acquisition and interpretation criteria is essential when using FDG-PET for monitoring disease activity. Consistency in acquisition parameters and interpretation across different timepoints in the same patient is crucial to ensure reliable comparisons [64,66]. FDG-PET findings can be influenced by several factors, including the timing, dose, and duration of glucocorticoid therapy [72,84,85,86], as sensitivity for detecting vasculitis is high within the first three days but decreases markedly thereafter [72]. Additional confounders include glucose levels, patient BMI, disease duration, atherosclerosis [87,88], especially relevant in elderly GCA patients, and the timing of image acquisition [70]. Despite the increasing adoption of FDG-PET, harmonised international guidelines for acquisition and interpretation in LVV remain lacking [66].

### 4.2. Takayasu Arteritis

Follow-up in TAK remains complex due to the challenge of differentiating active inflammation from chronic vascular damage, which may persist without overt clinical symptoms. Accurate assessment is essential for disease management, enabling therapeutic tapering with the least risk of damage progression. However, the nonspecific nature of clinical symptoms and biomarkers complicates this distinction, as acute-phase reactants such as ESR and CRP may remain normal in up to 50% of patients with clinically active disease [89].

Multiple studies have reported an association between FDG uptake scores—such as the PETVAS, SUV, and TBR, using the vena cava or liver as denominator—and clinical disease activity measures, including the Physician Global Assessment (PGA), ITAS-2010, NIH/Kerr criteria, and DEI.TAK [90,91,92,93]. Among these, PETVAS has been shown to correlate with global inflammatory burden. Kang et al. reported that PETVAS outperformed SUVmax in detecting disease activity, arguing that SUVmax reflects only focal uptake, while PETVAS better captures overall vascular involvement [94]. Conversely, other studies identified the TBR as having the strongest association with clinical disease activity [66,95]. However, Arnaud et al. found no correlation between visual scores or the TBR (denominator liver SUV) and clinical, biologic, or MRI-based measures, highlighting variability in performance depending on scoring methods and cohorts [96]. Notably, low-grade FDG uptake has been observed in some patients during clinical remission [97]. The clinical significance of this finding remains unclear, particularly in TAK, where it appears to be less frequent than in GCA. This difference may be related to the typically younger age of TAK patients and their lower prevalence of atherosclerosis [70]. Proposed cut-off values for SUVmax in defining active disease have ranged from 1.3 to 2.1 [98,99]. Tezuka et al. demonstrated that a threshold of 2.1 yielded a sensitivity of 92.6% and specificity of 91.7% for active disease, with higher diagnostic accuracy than CRP and ESR [99]. Nonetheless, Santhosh et al. cautioned that no universal cut-off can be established, recommending instead serial intra-patient comparisons to assess disease trends over time [100].

PET-derived metrics such as the SUV and TBR have shown consistent reductions following the initiation of immunosuppressive therapy, especially in patients with active disease at baseline [66,78,95,98,100,101,102]. Conversely, stability in uptake values has been reported in patients who did not undergo treatment modification. These findings underscore the potential of FDG-PET/CT as a dynamic tool for monitoring disease activity and therapeutic response [66]. The potential role of FDG-PET/CT in disease monitoring is further underscored by its integration into composite indices such as the Takayasu’s Arteritis Integrated Disease Activity Index (TAIDAI), which incorporates clinical features, serological markers, and imaging findings. In its development study, TAIDAI demonstrated excellent sensitivity (96.3%) and good specificity (79.2%) when compared with physician global assessment, and showed strong correlations with PETVAS, PGA, and inflammatory biomarkers [89].

FDG-PET/CT may also hold prognostic value. In a prospective study, Quinn et al. observed that most angiographic complications occurred in arterial territories with baseline FDG uptake (see Figure 5). Although the presence of FDG uptake at baseline did not independently predict angiographic progression, its absence was strongly associated with disease stability, with a negative predictive value of 99%. However, baseline PET findings were less closely associated with vascular progression than with clinical disease activity [103]. Furthermore, in a retrospective study, Tahra et al. found no significant difference in PETVASs between patients who later relapsed and those who remained in remission, emphasising the current limitations of FDG-PET for predicting future disease evolution [97]. In light of these limitations, as FDG uptake primarily reflects glycolytically active, macrophage-rich inflammation and may yield false-negative results in lesions dominated by other cell types such as fibroblasts, alternative tracers are being explored. Among these, fibroblast activation protein inhibitors (FAPI) and somatostatin receptor ligands (e.g., ^68^Ga-DOTATATE) have shown promising preliminary results. Nevertheless, their clinical applicability in LVV remains to be validated, and current evidence is still limited [104,105,106,107].

## 5. Computed Tomography (CT)/CT Angiography

CT and CT Angiography (CTA) are widely used in the assessment of LVV due to their broad availability, rapid acquisition time, and excellent spatial resolution. These modalities are particularly effective for detecting structural abnormalities such as luminal stenosis, occlusion, aneurysms, or vessel wall thickening. However, compared to metabolic imaging techniques like FDG-PET/CT or tissue-sensitive modalities such as MRI/MRA, CT and CTA provide limited information on active vessel wall inflammation. As such, their value lies predominantly in anatomical characterisation and the evaluation of vascular damage, rather than in the assessment of inflammatory activity [18,108,109].

The main disadvantages of CT-based imaging include ionising radiation and the need for iodinated contrast, both of which are particularly relevant in younger patients with TAK who may require repeated follow-up imaging. Consequently, CT and CTA are generally not recommended as first-line modalities for monitoring disease activity, but they play an important role in detecting complications and informing long-term management strategies [18].

### 5.1. Giant Cell Arteritis

In GCA, CTA does not play a role in the evaluation of cranial arteries, where ultrasound and MRI are preferred due to superior soft tissue resolution and absence of ionising radiation [18,110]. However, CTA may be employed to evaluate extracranial large vessel involvement when ultrasound, PET, or MRI are unavailable, though prospective evidence supporting its effectiveness during follow-up remains limited [18,111]. Characteristic features include mural thickening and double-ring enhancement during the venous phase [108]. In a prospective cohort of 40 patients with biopsy-proven GCA, contrast enhancement on CTA markedly decreased following glucocorticoid therapy; however, mural thickening persisted in two-thirds of cases after one year, underscoring the limited value of CTA for monitoring disease activity [109].

Due to its limitations in diagnosing active disease, CTA is not recommended for monitoring disease activity nor for investigating suspected relapses [18]. Nevertheless, it may be considered for long-term monitoring of structural damage, particularly at sites of previous vascular inflammation [2,87]. These structural changes, such as stenoses, occlusions, and arterial dilatations or aneurysms, can persist beyond clinical remission and impact long-term outcomes [18,112,113,114]. Despite this, routine baseline CTA for all patients is not advised, given concerns regarding radiation exposure, cost-effectiveness, and uncertain long-term benefit [18,59,115,116]. Instead, CTA may be selectively used in patients with signs or symptoms of vascular damage or recurrent/persistent inflammation involving large arteries such as the aorta (see Figure 6) [82,83].

### 5.2. Takayasu Arteritis

In TAK, CTA serves as an alternative to MRI—recommended as the first-line modality—and to FDG-PET for the evaluation of large vessel involvement [18].

The ability of CTA to detect treatment-related changes has been inconsistently reported [117,118]. In a retrospective observational study, Paul et al. evaluated 16 patients with early-phase TAK using serial electron beam CT. Vascular lesions progressed in 6 patients, while mural thickening decreased in more than half of follow-up scans. However, 25% of patients showed increased thickening despite treatment, reflecting the variable course of vascular inflammation and the potential value of CT in tracking structural changes [117]. In contrast, a retrospective study by Yoshida et al., focusing on supra-aortic branches, observed no consistent reduction in wall thickening following glucocorticoid therapy, possibly reflecting the limited sensitivity of CTA in detecting subtle changes in smaller-calibre vessels [118].

Similarly, studies evaluating the association between CTA-based vascular changes and inflammatory markers have yielded mixed results. In a prospective cross-sectional study, Sarma et al. examined 36 TAK patients using both CTA and MRA. Imaging-based activity strongly correlated with elevated ESR and CRP, present in 100% (12/12) and 91.6% (11/12) of patients with active lesions, respectively [119]. Conversely, in a retrospective cohort study, Keleşoğlu Dinçer et al. evaluated 97 TAK patients (53 with follow-up imaging) and found no significant association between radiological progression on CTA/MRA and changes in acute-phase reactants. Notably, over 70% of patients had normal ESR and/or CRP levels at follow-up, including 8 of the 15 patients with radiologic progression, highlighting the limited reliability of systemic inflammatory markers in reflecting ongoing vascular changes [120].

These discrepancies underscore the need for prospective, multimodal studies to better define the role of CTA in monitoring disease activity. Nevertheless, CTA remains valuable for detecting and following structural complications which can progress independently of inflammation (see Figure 7). To support standardised assessment of vascular damage, Tombetti et al. developed an already mentioned angiographic scoring system applicable to both CTA and MRA. Although MRA was more frequently used in their cohort—which also included a minority of GCA patients—the score was primarily validated in TAK. It quantifies stenoses and dilatations across 17 arterial territories. TAK patients exhibited higher stenotic and composite scores than GCA patients, reflecting distinct disease phenotypes. The score correlated with ITAS and patient global assessment and may serve as a valuable outcome measure in clinical trials and structured follow-up [59].

## 6. Conventional Angiography

Conventional angiography (CA), or catheter-based digital subtraction angiography, involves intra-arterial injection of contrast with fluoroscopic imaging, providing high-resolution views of the vascular lumen [121,122]. It also enables direct haemodynamic assessments, such as pressure gradients [123,124,125]. Historically central in the evaluation of LVV, CA has significant limitations: it is invasive, exposes patients to radiation and contrast, and does not allow for an assessment of the vessel wall, which is essential for evaluating active inflammation [124,125]. Procedural complications may include haematomas, pseudoaneurysms and dissections [121,126].

Over time, CA has been largely replaced by non-invasive imaging modalities—including CT angiography (CTA), MR angiography (MRA), and PET—which are now preferred for longitudinal assessment in both GCA and TAK. Its current use is mostly confined to guiding or planning endovascular or surgical interventions [18].

### 6.1. Giant Cell Arteritis

CA has limited utility in GCA. It is not suitable for assessing cranial arteries, and with the growing recognition of extracranial involvement, non-invasive modalities like ultrasound, CTA, MRA, and PET have become the standard for monitoring [62,127,128,129].

Nevertheless, CA may still be considered in selected cases involving vascular complications, such as aortic aneurysms, dissections, or stenoses, which may occur in up to 27% of patients during follow-up [113,115,129,130,131], particularly when detailed luminal mapping is required for interventional procedures.

### 6.2. Takayasu Arteritis

CA is no longer routinely used for monitoring TAK, having been largely replaced by newer imaging modalities that allow for repeated follow-up and provide better assessment of inflammatory activity without procedural risks [18,120,132,133].

Its current role is primarily confined to evaluating chronic vascular sequelae—such as occlusions or aneurysms—and to guiding interventions like angioplasty or stenting [134,135,136]. While CA findings contributed historically to disease activity scores such as the NIH/Kerr and ITAS2010 criteria [91,137], its inability to visualise vessel wall inflammation limits its relevance in current clinical practice. Retrospective studies have documented lesion progression on CA even in patients in clinical remission [132,137,138], but such findings are now more appropriately captured using non-invasive imaging.

## 7. Multimodal Assessment

Different imaging modalities offer distinct yet complementary strengths, and their integration can enhance the interpretation of disease activity and vascular damage (Table 1). This multimodal approach is particularly valuable when multiple explanations are possible—such as differentiating vascular remodelling, atherosclerosis, and active inflammation—or when clinical and laboratory findings are inconclusive, as often occurs in patients receiving IL-6 blockade. Imaging selection should be guided by the clinical question, vascular territory involved, and disease phenotype.

According to the EULAR 2023 recommendations, ultrasound is the first-line imaging test for suspected GCA [18]. When the result is negative or inconclusive but clinical suspicion remains high, additional imaging and/or histology is advised. The choice of the next investigation should be guided by whether cranial or extracranial involvement is suspected, local availability, and pre-test probability. Head-to-head data support this phenotype-guided, sequential strategy. In the prospective comparison by van Nieuwland et al. [139], all patients with suspected GCA underwent ultrasound, whole-body FDG-PET/CT, and cranial MRI within five days. Sensitivity was 69.6% for ultrasound, 52.2% for FDG-PET/CT, and 56.5% for MRI, with 100% specificity for all modalities. Importantly, FDG-PET/CT was negative in all isolated cranial GCA cases, while cranial MRI was negative in all isolated extracranial GCA cases. In several patients with negative or inconclusive ultrasound, subsequent imaging confirmed the diagnosis, reinforcing that ultrasound can serve as the first test and be complemented by a phenotype-guided second modality when diagnostic uncertainty persists. This principle has been operationalised in a large real-world ultrasonography-led multimodal pathway reported by Mukhtyar et al. in 1000 consecutive referrals [140]. The initial step consisted of scanning the superficial temporal and axillary arteries, where ultrasound positivity required involvement of ≥two arteries (unilateral temporal-artery disease alone was considered negative, prompting extended scanning). If the initial ultrasound was negative and CRP ≥ 20 mg/L, a manifestation-based extended ultrasound of additional cranial or extracranial branches was performed. When no vasculitis was found and no alternative diagnosis was evident, a second test was obtained—temporal-artery biopsy for cranial-predominant symptoms or FDG-PET/CT for constitutional/extracranial presentations. Diagnostic performance improved stepwise: 78.7% sensitivity for temporal + axillary scanning, 86.1% after extended ultrasound, and 95.7% after including a second test, with a negative predictive value of 98.0%. For comparison, temporal-artery imaging alone achieved 62.3% sensitivity. This structured, multimodal algorithm allowed for safe glucocorticoid tapering when results remained negative.

Beyond diagnosis, a multimodal strategy also informs relapse monitoring. Ultrasound has shown moderate accuracy in longitudinal cohorts, and dynamic changes in ultrasound activity scores may predict relapse. Cranial MRI can reveal vascular changes not mirrored by clinical symptoms or inflammatory markers, indicating that the modalities capture different aspects of disease biology [141]. Selective use of additional imaging at suspected relapse—rather than routine serial scanning in stable remission—aligns with EULAR recommendations for pragmatic, phenotype-guided follow-up [18].

Hybrid imaging techniques such as PET/MRI, which integrate metabolic and structural data, may further enhance diagnostic precision. In this context, multimodal quantitative tools such as the Vasculitis Activity using MR-PET (VAMP) score have been proposed to standardise disease-activity assessment by combining MRI vessel-wall enhancement with PET metabolic activity to provide a reproducible, objective index of inflammation [142,143]. Together, these advances underscore the potential of multimodal imaging to improve diagnostic confidence, precision, and reproducibility in the longitudinal management of LVV.

## 8. Future Perspectives

Emerging technologies hold considerable promise for refining the monitoring of LVV. Artificial intelligence (AI) and machine learning are poised to enable automated assessment of vessel wall changes, thereby improving reproducibility and reducing interobserver variability. Recent proof-of-concept studies have already explored these applications. Vries et al. demonstrated a radiomics-based machine learning model that accurately distinguished active GCA from atherosclerosis in the aorta on follow-up [18F]FDG-PET/CT scans [144], while Roncato et al. applied a U-Net convolutional neural network to detect the halo sign in temporal artery ultrasound images, assisting GCA diagnosis [145]. These approaches illustrate how AI can already support image interpretation and are likely to extend into disease activity monitoring and longitudinal follow-up.

The role of imaging in guiding treatment tapering or escalation, however, remains uncertain. To date, no study has directly compared conventional monitoring—based on clinical and laboratory parameters—with integrated clinical–laboratory–imaging strategies to demonstrate improved outcomes. This evidence gap helps explain why the 2023 EULAR recommendations advise against routine serial imaging in the absence of clinical suspicion of relapse [18].

Although a universally accepted imaging ‘gold standard’ for defining relapse in LVV is still lacking, validated indices are emerging. In GCA, the OGUS has demonstrated high reliability, sensitivity to change, and growing prognostic value across multiple cohorts. In TAK, the composite TAIDAI—integrating clinical, laboratory, and imaging domains—has shown excellent sensitivity and good specificity in development studies. Further external validation, threshold harmonisation, and linkage to patient-centred outcomes remain priorities before protocolised imaging-guided treat-to-target approaches can be widely recommended. Incorporating imaging biomarkers into treat-to-target frameworks may ultimately enable more personalised, proactive, and outcome-focused management of LVV.

## 9. Conclusions

Imaging plays a crucial role in the monitoring of large vessel vasculitis. While ultrasound remains a practical tool for assessing superficial arteries in GCA, MRI, PET, CT, and CA each provide complementary insights into disease activity and vascular damage. A multimodal approach, tailored to individual clinical scenarios, is often required.

Despite notable progress, there is a need for greater standardisation and integration of imaging into routine clinical practice, as the clinical significance of subtle or residual vascular changes is still uncertain, and treatment decisions should not rely on these findings alone. Future research should aim to validate imaging-based outcome measures harmonise acquisition and interpretation protocols to enhance the accuracy and reproducibility of monitoring across centres. With ongoing innovation, imaging will continue to play a pivotal role in advancing the management of LVV.

## Figures and Tables

**Figure 1 biomolecules-15-01505-f001:**
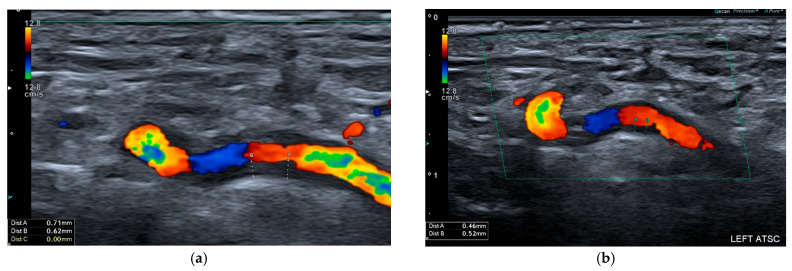
Ultrasound of the left proximal temporal artery, in longitudinal view, depicting a halo sign in a patient with GCA, before (**a**) and after (**b**) 3 months of glucocorticoid treatment.

**Figure 2 biomolecules-15-01505-f002:**
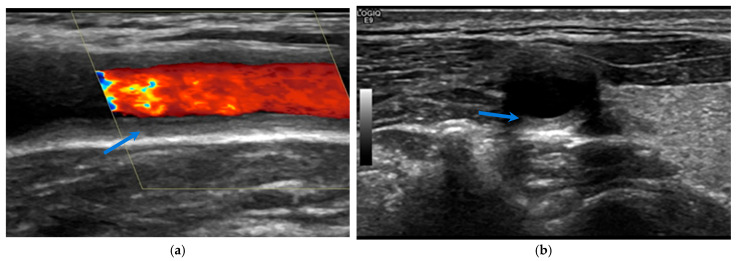
Ultrasound of the right carotid, in longitudinal (**a**) and transverse (**b**) views, showing a macaroni sign (blue arrows) in a 19-year-old patient with Takayasu arteritis.

**Figure 3 biomolecules-15-01505-f003:**
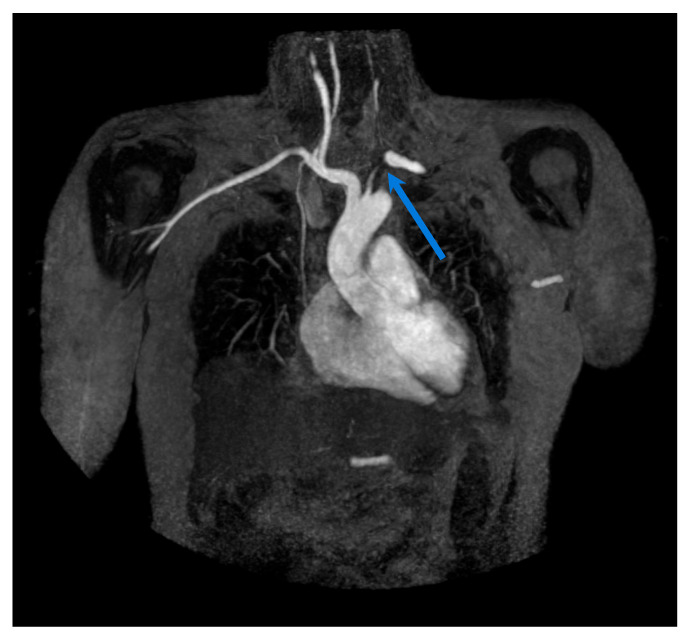
MRA (T2-weighted) in a patient with Takayasu arteritis showing absent flow in the left subclavian artery (blue arrow) and inflammatory wall thickening of the left common carotid artery with >50% proximal stenosis.

**Figure 4 biomolecules-15-01505-f004:**
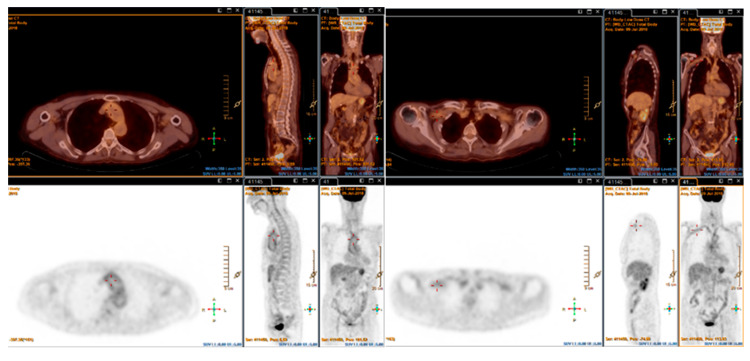
Multiplanar fused FDG-PET/CT showing vascular hypermetabolism in the subclavian arteries, thoracic aorta (including the arch), and femoral arteries in a patient with GCA (SUVmax 3.4).

**Figure 5 biomolecules-15-01505-f005:**
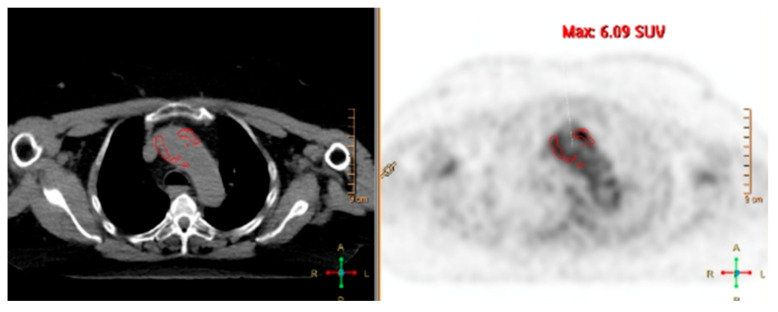
Axial fused FDG-PET/CT from a patient with Takayasu arteritis, demonstrating intense vascular hypermetabolism in the aortic arch (SUVmax 6.09).

**Figure 6 biomolecules-15-01505-f006:**
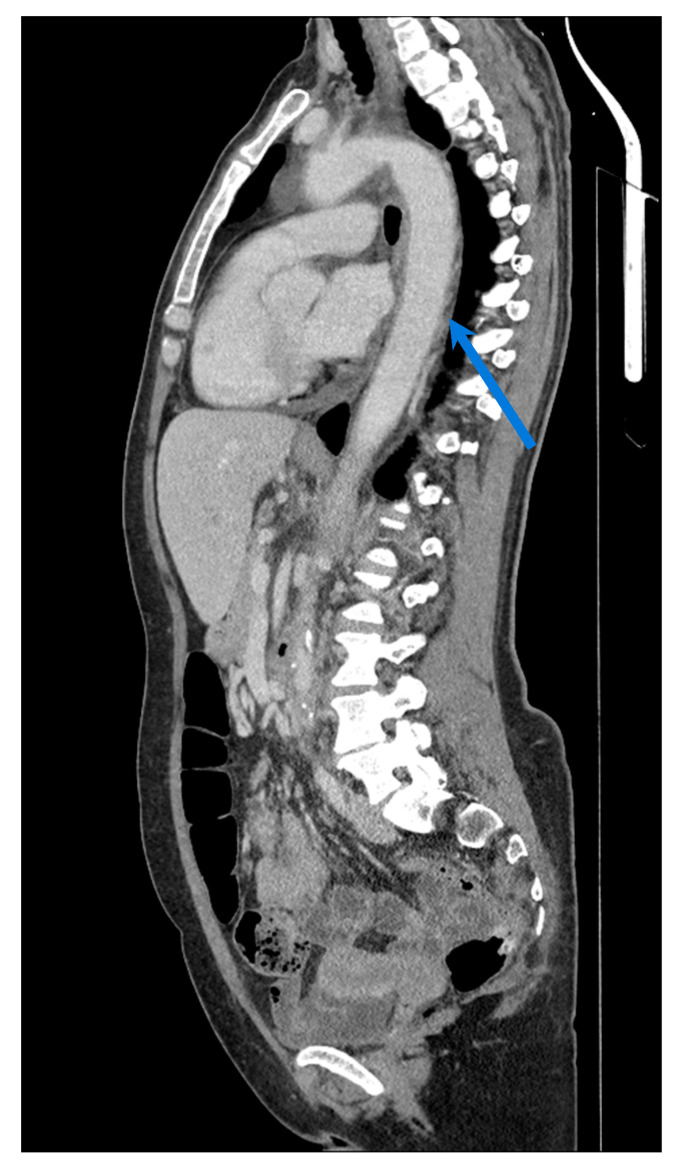
Angio-CT showing wall thickening (blue arrow) of the thoracic aorta in a patient with GCA.

**Figure 7 biomolecules-15-01505-f007:**
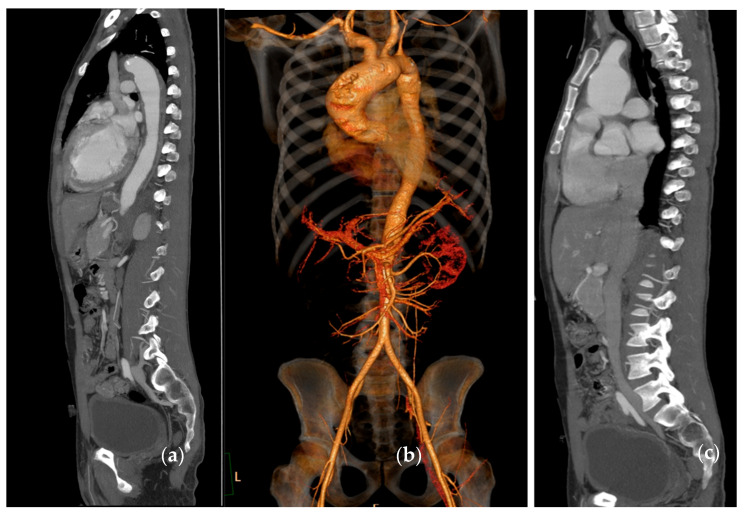
Angio-CT showing an ascending aortic aneurysm (5.1 cm) with aortic insufficiency in a 28-year-old male with Takayasu arteritis (same patient seen in Figure 5). Panel (**a**) shows a sagittal view showing wall thickening in the thoracic aorta; (**b**) 3D volume-rendered reconstruction of the CT angiogram; (**c**) a sagittal view of the aneurysm.

**Table 1 biomolecules-15-01505-t001:** Comparative features of imaging modalities in large vessel vasculitis.

Modality	Resolution	Specificity	Monitoring Capability	Clinical Utility	Limitations
Ultrasound	High for superficial arteries; limited for deep vessels.	Moderate–high, mostly based on halo and macaroni signs.	Sensitive to inflammatory changes (wall thickness and halo sign). Responsive under IL-6 blockade. Enables evaluation of structural damage.	Non-invasive, radiation-free, cost-effective, repeatable.	Operator-dependent; limited resolution for deep vessel evaluation; uncertain significance of residual or subclinical vascular changes.
MRI/MRA	High for vessel wall and lumen evaluation; broad anatomic coverage.	High—detects mural thickening, oedema, and contrast enhancement.	Suitable for serial monitoring of inflammation and structural damage.	Radiation-free. Preferred imaging monitoring relapse and disease extent in TAK.	Limited availability and high cost; contraindicated in patients with non-MRI-compatible devices. The clinical significance of persistent mural enhancement or subclinical findings remains uncertain.
FDG-PET/CT	Moderate; whole-body coverage.	High for active metabolic activity.	Sensitive to metabolic changes; useful for assessing treatment response.	Valuable in detecting relapse and for patients receiving IL-6 inhibitors.	Radiation exposure, high cost, limited availability, and low spatial resolution; limited sensitivity for vascular complications; uncertain significance of persistent or low-grade vascular uptake.
CT/CTA	High; wide anatomic coverage.	High for vascular structural characterisation; poor for inflammatory changes.	Better suited for evaluating chronic vascular damage rather than active inflammation.	Widely available; rapid acquisition time; enables anatomic mapping.	Ionising radiation; use of iodinated contrast; limited ability to detect active inflammation.
Conventional angiography	Moderate.	High for vascular lumen characterisation; no wall assessment.	Useful for evaluating structural vessel damage; not suitable for inflammation monitoring.	Reserved for guiding or planning endovascular interventions.	Invasive; radiation exposure; contrast use; procedural risks.

CT—Computed Tomography; CTA—Computed Tomography Angiography; FDG—PET Fluorodeoxyglucose Positron Emission Tomography; GCA—Giant Cell Arteritis; IL-6—Interleukin-6; MRA—Magnetic Resonance Angiography; MRI—Magnetic Resonance Imaging; TAK—Takayasu Arteritis.

## Data Availability

Data sharing is not applicable.

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
