# Peer review of "The Role of Imaging in Monitoring Large Vessel Vasculitis: A Comprehensive Review"

_biomolecules, 2025, doi:10.3390/biom15111505_

Round 1

Reviewer 1 Report

Comments and Suggestions for Authors
  1. The absence of a defined methodology—theirs lacks inclusion/exclusion criteria, database sources, and a time frame for literature selection—makes the review strengthen no informational contribution whatsoever.
  2. There is little discussion on the limitations of each modality, such as false positives/negatives, cost, and accessibility.
  3. Newer indices such as OMERACT and TAIDAI are proposed without proper further discussion or gold standard comparisons.
  4. A comparison table of imaging modalities categorized by resolution, specificity, ability to monitor the disease and clinical utility is absent.
  5. There are no workarounds for IL-6 receiver blocker patients regarding imaging challenges—the review does not adequately elaborate on specific false-negative scenarios.
  6. However, the role of hybrid imaging (e.g., PET/MRI) and machine learning-based quantification methods is not explored, although they are quite relevant in the contemporary practice and research.
  7. Lacks discussion on the use of imaging in directing tapering or escaltion of immunosuppressive therapy—limits clinical applicability
  8. Broader clinical audiences will find the scope limited by the absence of pediatric or atypical LVV presentations.
  9. Readers encounter frequent references to a “multimodal approach”, but never its operationalization – no one really knows what this looks like in the field.
  10. And recent imaging studies (2022–2024) of disease activity scoring and relapse prediction in GCA and TAKhave been missed in update citation to conclude, GCA and TAK are rare vasculitis conditions best managed by imaging guided.
  • Tao X, Wang J, Yan Y, Cheng P, Liu B, Du H and Niu B (2025) Optimal Sca-1-based procedure for purifying mouse adipose-derived mesenchymal stem cells with enhanced proliferative and differentiation potential. Front. Cell Dev. Biol. 13:1566670. doi: 10.3389/fcell.2025.1566670
  • Wang, H., He, Y., Wan, L., Li, C., Li, Z., Li, Z.,... Tu, C. (2025). Deep learning models in classifying primary bone tumors and bone infections based on radiographs. npj Precision Oncology, 9(1), 72. doi: 10.1038/s41698-025-00855-3
  • Yilin, Z., Haiquan, F., Chen, H., & Juan, S. (2025). Hemodynamics of asymmetrically stenotic vertebral arteries based on fluid–solid coupling. Journal of Biological Physics, 51(1), 10. doi: 10.1007/s10867-025-09673-x

All references mentioned are suggested but not mandatory.

Comments on the Quality of English Language

The English could be improved to more clearly express the research.

Author Response

We would like to express our gratitude for the invitation extended to Prof. Cristina Ponte to submit a revision paper on the role of imaging in monitoring disease activity in large vessel vasculitis. We are writing to resubmit the revised version of our manuscript for consideration for publication in Biomolecules.

We appreciate the opportunity to revise our work based on the insightful feedback received from the reviewers, and we believe that our revised manuscript now addresses the concerns raised during the initial evaluation process.

We have carefully considered the comments of the reviewers and prepared a point-by-point response to address each of the reviewers’ concerns.

Reviewer 1:

  1. The absence of a defined methodology—theirs lacks inclusion/exclusion criteria, database sources, and a time frame for literature selection—makes the review strengthen no informational contribution whatsoever.

Reply: We thank the reviewer for this observation. We would like to clarify that our manuscript is a narrative review; therefore, a formal search strategy, such as those typically required in systematic reviews, was not part of our methodology. Narrative reviews are inherently more flexible and are designed to provide a broad overview of a topic based on expert knowledge and a comprehensive understanding of the available literature. While we did not include a formal “Literature Search” section, we have thoroughly reviewed the relevant studies and key findings to support our conclusions. To make this clearer we have added in the 4th sentence of the abstract: “This narrative review provides a comprehensive overview of the role of imaging in monitoring LVV.” and, in the Introduction section (Section 1, 3rd paragraph): “This narrative review provides a comprehensive overview of the role of imaging in monitoring large vessel vasculitis”.

  1. There is little discussion on the limitations of each modality, such as false positives/negatives, cost, and accessibility.

Reply: We thank the reviewer for this valuable remark. In response, and considering related suggestions, we have revised the text in Section 2 (1st paragraph) to better highlight these aspects. We now emphasize that: “Although these sonographic signs are highly characteristic of vascular inflammation, false positives can occur, most often due to overlap with atherosclerotic changes [4]. This underscores the importance of operator expertise and standardized protocols, while also highlighting the increased cardiovascular burden in LVV patients, in whom accelerated atheromatosis and arteriosclerosis contribute to excess morbidity and mortality [5].” 

Similarly, in the Fluorine-18-fluorodeoxyglucose PET Section (Section 4.2, last paragraph), we clarified the following: “In light of these limitations, as FDG uptake primarily reflects glycolytically active, macrophage-rich inflammation and may yield false-negative results in lesions dominated by other cell types such as fibroblasts, alternative tracers are being explored. Among these, fibroblast activation protein inhibitors (FAPI) and somatostatin receptor ligands (e.g., ^68Ga-DOTATATE) have shown promising preliminary results. Nevertheless, their clinical applicability in LVV remains to be validated, and current evidence is still limited [103-106].”

Additionally, we have added a new Section 7 (Multimodal assessment), which includes a table summarising the key characteristics of each imaging modality, outlining their respective strengths and limitations.

Table 1. Comparative features of imaging modalities in large vessel vasculitis.

Modality

Resolution

Specificity

Monitoring capability

Clinical utility

Limitations

Ultrasound

High for superficial arteries;

limited for deep vessels.

Moderate-high, mostly based on halo and macaroni signs.

Sensitive to inflammatory changes (wall thickness and halo sign).

Responsive under IL-6 blockade.

Enables evaluation of structural damage.

Non-invasive, radiation-free, cost-effective, repeatable.

Operator-dependent; limited resolution for deep vessel evaluation;

uncertain significance of residual or subclinical vascular changes.

MRI/MRA

High for vessel wall and lumen evaluation;

broad anatomic coverage.

High-detects mural thickening, oedema, and contrast enhancement.

Suitable for serial monitoring of inflammation and structural damage.

Radiation-free.

Preferred imaging monitoring relapse and disease extent in TAK.

 Limited availability and high cost; contraindicated in patients with non-MRI-compatible devices.

 The clinical significance of persistent mural enhancement or subclinical findings remains uncertain.

FDG-PET/CT

Moderate;

whole-body coverage.

High for active metabolic activity.

Sensitive to metabolic changes; useful for assessing treatment response.

Valuable in detecting relapse and for patients receiving IL-6 inhibitors.

Radiation exposure, high cost, limited availability, and low spatial resolution; limited sensitivity for vascular complications; uncertain significance of persistent or low-grade vascular uptake.

CT/CTA

High; wide anatomic
coverage.

High for vascular structural characterisation; poor for inflammatory changes.

Better suited for evaluating chronic vascular damage rather than active inflammation.

Widely available; rapid acquisition time; enables anatomic mapping.

Ionising radiation; use of iodinated contrast; limited ability to detect active inflammation.

Conventional angiography

Moderate.

High for vascular lumen characterisation; no wall assessment.

Useful for evaluating structural vessel damage; not suitable for inflammation monitoring.

Reserved for guiding or planning endovascular interventions.

Invasive; radiation exposure; contrast use; procedural risks.

CT – Computed Tomography; CTA – Computed Tomography Angiography; FDG – PET Fluorodeoxyglucose Positron Emission Tomography; GCA – Giant Cell Arteritis; IL-6 – Interleukin-6; MRA – Magnetic Resonance Angiography; MRI – Magnetic Resonance Imaging; TAK – Takayasu Arteritis

  1. Newer indices such as OMERACT and TAIDAI are proposed without proper further discussion or gold standard comparisons.

Reply: We thank the reviewer for this valuable comment. Although both indices offer structured and reproducible frameworks for imaging assessment, neither has yet been validated against a definitive gold standard. Further studies are therefore needed to establish clinical correlation and outcome prediction. To address this point, we have added the following clarification in the Section 8 (Future Perspectives; 3rd paragraph): “Although a universally accepted imaging ‘gold standard’ for defining relapse in LVV is still lacking, validated indices are emerging. In GCA, the OGUS has demonstrated high reliability, sensitivity to change, and growing prognostic value across multiple cohorts. In TAK, the composite TAIDAI index - integrating clinical, laboratory, and imaging domains - has shown excellent sensitivity and good specificity in development studies. Further external validation, threshold harmonisation, and linkage to patient-centred outcomes remain priorities before protocolised imaging-guided treat-to-target approaches can be widely recommended.”

  1. A comparison table of imaging modalities categorized by resolution, specificity, ability to monitor the disease and clinical utility is absent.

Reply: We thank the reviewer for this valuable suggestion. In response, we have added a new summary table (Table 1) providing a comparative overview of the main imaging modalities, categorised by resolution, specificity, monitoring capability, clinical utility, and limitations. This table has been placed after the detailed discussion of each imaging technique to summarise key features and facilitate direct comparison across modalities.

Table 1. Comparative features of imaging modalities in large vessel vasculitis.

Modality

Resolution

Specificity

Monitoring capability

Clinical utility

Limitations

Ultrasound

High for superficial arteries;

limited for deep vessels.

Moderate-high, mostly based on halo and macaroni signs.

Sensitive to inflammatory changes (wall thickness and halo sign).

Responsive under IL-6 blockade.

Enables evaluation of structural damage.

Non-invasive, radiation-free, cost-effective, repeatable.

Operator-dependent; limited resolution for deep vessel evaluation;

uncertain significance of residual or subclinical vascular changes.

MRI/MRA

High for vessel wall and lumen evaluation;

broad anatomic coverage.

High-detects mural thickening, oedema, and contrast enhancement.

Suitable for serial monitoring of inflammation and structural damage.

Radiation-free.

Preferred imaging monitoring relapse and disease extent in TAK.

 Limited availability and high cost; contraindicated in patients with non-MRI-compatible devices.

 The clinical significance of persistent mural enhancement or subclinical findings remains uncertain.

FDG-PET/CT

Moderate;

whole-body coverage.

High for active metabolic activity.

Sensitive to metabolic changes; useful for assessing treatment response.

Valuable in detecting relapse and for patients receiving IL-6 inhibitors.

Radiation exposure, high cost, limited availability, and low spatial resolution; limited sensitivity for vascular complications; uncertain significance of persistent or low-grade vascular uptake.

CT/CTA

High; wide anatomic
coverage.

High for vascular structural characterisation; poor for inflammatory changes.

Better suited for evaluating chronic vascular damage rather than active inflammation.

Widely available; rapid acquisition time; enables anatomic mapping.

Ionising radiation; use of iodinated contrast; limited ability to detect active inflammation.

Conventional angiography

Moderate.

High for vascular lumen characterisation; no wall assessment.

Useful for evaluating structural vessel damage; not suitable for inflammation monitoring.

Reserved for guiding or planning endovascular interventions.

Invasive; radiation exposure; contrast use; procedural risks.

CT – Computed Tomography; CTA – Computed Tomography Angiography; FDG – PET Fluorodeoxyglucose Positron Emission Tomography; GCA – Giant Cell Arteritis; IL-6 – Interleukin-6; MRA – Magnetic Resonance Angiography; MRI – Magnetic Resonance Imaging; TAK – Takayasu Arteritis

  1. There are no workarounds for IL-6 receiver blocker patients regarding imaging challenges—the review does not adequately elaborate on specific false-negative scenarios.

Reply: We thank the reviewer for highlighting this important point. We have expanded the discussion to better address imaging challenges in patients receiving IL-6 inhibitors. In the Ultrasound section (Section 2, subsection 2.1, third paragraph), we now specify that: “Ultrasound remains informative in patients treated with IL-6 inhibitors, in whom suppression of systemic inflammatory markers limits the reliability of laboratory monitoring. Seitz et al. and Sebastian et al. showed that findings such as halo persistence and changes in IMT remain responsive despite low CRP and ESR levels, thereby providing an objective measure of disease activity [6,8]. This makes ultrasound particularly valuable for relapse detection when laboratory results are inconclusive. It should be noted, however, that not all relapses are accompanied by vascular changes—those manifesting with isolated polymyalgia rheumatica (PMR) symptoms, for instance, may occur without detectable sonographic abnormalities. Such relapses are generally considered less severe and are unlikely to reflect active vascular inflammation [13].”. In addition, this concept is reinforced in the new Section 7 (Multimodal assessment, first paragraph): “Different imaging modalities offer distinct yet complementary strengths, and their integration can enhance the interpretation of disease activity and vascular damage (Table 1). This multimodal approach is particularly valuable when multiple explanations are possible—such as differentiating vascular remodelling, atherosclerosis, and active inflammation—or when clinical and laboratory findings are inconclusive, as often occurs in patients receiving IL-6 blockade. Imaging selection should be guided by the clinical question, vascular territory involved, and disease phenotype.”

  1. However, the role of hybrid imaging (e.g., PET/MRI) and machine learning-based quantification methods is not explored, although they are quite relevant in the contemporary practice and research.

Reply: We thank the reviewer for this insightful comment. In response, and in line with other suggestions, we have created a new Section 7 (Multimodal assessment), where we discuss the potential of multimodal and hybrid imaging approaches in large-vessel vasculitis. The section now reads:

“Different imaging modalities offer distinct yet complementary strengths, and their integration can enhance the interpretation of disease activity and vascular damage (Table 1). This multimodal approach is particularly valuable when multiple explanations are possible—such as differentiating vascular remodelling, atherosclerosis, and active inflammation—or when clinical and laboratory findings are inconclusive, as often occurs in patients receiving IL-6 blockade. Imaging selection should be guided by the clinical question, vascular territory involved, and disease phenotype.

According to the EULAR 2023 recommendations, ultrasound is the first-line imaging test for suspected giant cell arteritis (GCA) [18]. When the result is negative or inconclusive but clinical suspicion remains high, additional imaging and/or histology is advised. The choice of the next investigation should be guided by whether cranial or extracranial involvement is suspected, local availability, and pre-test probability. Head-to-head data support this phenotype-guided, sequential strategy. In the prospective comparison by van Nieuwland et al. [139], all patients with suspected GCA underwent ultrasound, whole-body FDG-PET/CT, and cranial MRI within five days. Sensitivity was 69.6% for ultrasound, 52.2% for FDG-PET/CT, and 56.5% for MRI, with 100% specificity for all modalities. Importantly, FDG-PET/CT was negative in all isolated cranial GCA cases, while cranial MRI was negative in all isolated extracranial GCA cases. In several patients with negative or inconclusive ultrasound, subsequent imaging confirmed the diagnosis, reinforcing that ultrasound can serve as the first test and be complemented by a phenotype-guided second modality when diagnostic uncertainty persists. This principle has been operationalised in a large real-world ultrasonography-led multimodal pathway reported by Mukhtyar et al. in 1,000 consecutive referrals [140]. The initial step consisted of scanning the superficial temporal and axillary arteries, where ultrasound positivity required involvement of ≥ two arteries (unilateral temporal-artery disease alone was considered negative, prompting extended scanning). If initial ultrasound was negative and CRP ≥ 20 mg/L, a manifestation-based extended ultrasound of additional cranial or extracranial branches was performed. When no vasculitis was found and no alternative diagnosis was evident, a second test was obtained—temporal-artery biopsy for cranial-predominant symptoms or FDG-PET/CT for constitutional/extracranial presentations. Diagnostic performance improved stepwise: 78.7% sensitivity for temporal + axillary scanning, 86.1% after extended ultrasound, and 95.7% after including a second test, with a negative predictive value of 98.0%. For comparison, temporal-artery imaging alone achieved 62.3% sensitivity. This structured, multimodal algorithm allowed safe glucocorticoid tapering when results remained negative.

Beyond diagnosis, a multimodal strategy also informs relapse monitoring. Ultrasound has shown moderate accuracy in longitudinal cohorts, and dynamic changes in ultrasound activity scores may predict relapse. Cranial MRI can reveal vascular changes not mirrored by clinical symptoms or inflammatory markers, indicating that the modalities capture different aspects of disease biology. Selective use of additional imaging at suspected relapse—rather than routine serial scanning in stable remission—aligns with EULAR recommendations for pragmatic, phenotype-guided follow-up [18].

Hybrid imaging techniques such as PET/MRI, which integrate metabolic and structural data, may further enhance diagnostic precision. In this context, multimodal quantitative tools such as the Vasculitis Activity using MR-PET (VAMP) score have been proposed to standardise disease-activity assessment by combining MRI vessel wall enhancement with PET metabolic activity to provide a reproducible, objective index of inflammation [142,143]. Together, these advances underscore the potential of multimodal imaging to improve diagnostic confidence, precision, and reproducibility in the longitudinal management of LVV.“

In addition, we have expanded Section 8 (Future perspectives, first paragraph) to address machine-learning-based quantification: “Emerging technologies hold considerable promise for refining the monitoring of LVV. Artificial intelligence (AI) and machine learning are poised to enable automated assessment of vessel wall changes, thereby improving reproducibility and reducing interobserver variability. Recent proof-of-concept studies have already explored these applications. Vries et al. demonstrated a radiomics-based machine learning model that accurately distinguished active GCA from atherosclerosis in the aorta on follow-up [18F]FDG-PET/CT scans [144], while Roncato et al. applied a U-Net convolutional neural network to detect the halo sign in temporal artery ultrasound images, assisting GCA diagnosis [145]. These approaches illustrate how AI can already support image interpretation and are likely to extend into disease activity monitoring and longitudinal follow-up”

  1. Lacks discussion on the use of imaging in directing tapering or escalation of immunosuppressive therapy—limits clinical applicability

Reply: We thank the reviewer for this insightful comment. We agree that the clinical applicability of imaging in guiding the tapering or escalation of immunosuppressive therapy is a crucial topic. To address this, we have expanded the Future Perspectives section (Section 8, second to fourth paragraphs) as follows: “The role of imaging in guiding treatment tapering or escalation, however, remains uncertain. To date, no study has directly compared conventional monitoring - based on clinical and laboratory parameters - with integrated clinical-laboratory-imaging strategies to demonstrate improved outcomes. This evidence gap helps explain why the 2023 EULAR recommendations advise against routine serial imaging in the absence of clinical suspicion of relapse [18].

Although a universally accepted imaging ‘gold standard’ for defining relapse in LVV is still lacking, validated indices are emerging. In GCA, the OGUS has demonstrated high reliability, sensitivity to change, and growing prognostic value across multiple cohorts. In TAK, the composite TAIDAI index - integrating clinical, laboratory, and imaging domains - has shown excellent sensitivity and good specificity in development studies. Further external validation, threshold harmonisation, and linkage to patient-centred outcomes remain priorities before protocolised imaging-guided treat-to-target approaches can be widely recommended.

Collectively, these advances point toward standardised, phenotype-guided, and data-driven imaging strategies that can be applied consistently across clinical settings. Incorporating imaging biomarkers into treat-to-target frameworks may ultimately enable more personalised, proactive, and outcome-focused management of LVV.”

We also added the following note in the Conclusions Section (Section 9, 2nd paragraph): “Despite notable progress, there is a need for greater standardisation and integration of imaging into routine clinical practice, as the clinical significance of subtle or residual vascular changes is still uncertain, and treatment decisions should not rely on these findings alone.”

  1. Broader clinical audiences will find the scope limited by the absence of pediatric or atypical LVV presentations.

Reply: We thank the reviewer for raising this important point. We agree that paediatric and atypical LVV presentations represent relevant and distinct clinical entities. However, the aim of this review was to provide a focused and detailed overview of imaging approaches for monitoring typical adult-onset GCA and TAK. Expanding the discussion to include paediatric or atypical forms would have substantially broadened the scope and length of the manuscript, and it would not have been feasible to include representative examples for each imaging modality while maintaining balanced coverage. To make this clearer, we have amended the Introduction (Section 1, third paragraph) to read: “The scope of this review is limited to adult-onset GCA and TAK and does not extend to paediatric or atypical presentations.”

  1. Readers encounter frequent references to a “multimodal approach”, but never its operationalization – no one really knows what this looks like in the field.

Reply:  We thank the reviewer for this insightful comment. We agree that the concept of a “multimodal approach” is frequently mentioned in the literature but often lacks practical definition. In response, we have expanded the manuscript with a new Section 7 (Multimodal assessment), which clarifies how different imaging modalities can be integrated into clinical decision-making and longitudinal monitoring. The section now explicitly describes how multimodal imaging may be applied in practice, in light of current recommendations and recent literature: “According to the EULAR 2023 recommendations, ultrasound is the first-line imaging test for suspected giant cell arteritis (GCA) [18]. When the result is negative or inconclusive but clinical suspicion remains high, additional imaging and/or histology is advised. The choice of the next investigation should be guided by whether cranial or extracranial involvement is suspected, local availability, and pre-test probability. Head-to-head data support this phenotype-guided, sequential strategy. In the prospective comparison by van Nieuwland et al. [139], all patients with suspected GCA underwent ultrasound, whole-body FDG-PET/CT, and cranial MRI within five days. Sensitivity was 69.6% for ultrasound, 52.2% for FDG-PET/CT, and 56.5% for MRI, with 100% specificity for all modalities. Importantly, FDG-PET/CT was negative in all isolated cranial GCA cases, while cranial MRI was negative in all isolated extracranial GCA cases. In several patients with negative or inconclusive ultrasound, subsequent imaging confirmed the diagnosis, reinforcing that ultrasound can serve as the first test and be complemented by a phenotype-guided second modality when diagnostic uncertainty persists. This principle has been operationalised in a large real-world ultrasonography-led multimodal pathway reported by Mukhtyar et al. in 1,000 consecutive referrals [140]. The initial step consisted of scanning the superficial temporal and axillary arteries, where ultrasound positivity required involvement of ≥ two arteries (unilateral temporal-artery disease alone was considered negative, prompting extended scanning). If initial ultrasound was negative and CRP ≥ 20 mg/L, a manifestation-based extended ultrasound of additional cranial or extracranial branches was performed. When no vasculitis was found and no alternative diagnosis was evident, a second test was obtained—temporal-artery biopsy for cranial-predominant symptoms or FDG-PET/CT for constitutional/extracranial presentations. Diagnostic performance improved stepwise: 78.7% sensitivity for temporal + axillary scanning, 86.1% after extended ultrasound, and 95.7% after including a second test, with a negative predictive value of 98.0%. For comparison, temporal-artery imaging alone achieved 62.3% sensitivity. This structured, multimodal algorithm allowed safe glucocorticoid tapering when results remained negative.

Beyond diagnosis, a multimodal strategy also informs relapse monitoring. Ultrasound has shown moderate accuracy in longitudinal cohorts, and dynamic changes in ultrasound activity scores may predict relapse. Cranial MRI can reveal vascular changes not mirrored by clinical symptoms or inflammatory markers, indicating that the modalities capture different aspects of disease biology. Selective use of additional imaging at suspected relapse—rather than routine serial scanning in stable remission—aligns with EULAR recommendations for pragmatic, phenotype-guided follow-up [18].”

  1. And recent imaging studies (2022–2024) of disease activity scoring and relapse prediction in GCA and TAK have been missed in update citation to conclude, GCA and TAK are rare vasculitis conditions best managed by imaging guided.

We thank the reviewer for this helpful suggestion and for highlighting the importance of incorporating the most recent literature. We agree that new imaging studies on scoring systems and relapse prediction in GCA and TAK are essential to strengthen the review; we have added this information in Fluorine-18-fluorodeoxyglucose PET Section (Section 4, subsection 4.2, 4th paragraph): “In light of these limitations, as FDG uptake primarily reflects glycolytically active, macrophage-rich inflammation and may yield false-negative results in lesions dominated by other cell types such as fibroblasts, alternative tracers are being explored. Among these, fibroblast activation protein inhibitors (FAPI) and somatostatin receptor ligands (e.g., ^68Ga-DOTATATE) have shown promising preliminary results. Nevertheless, their clinical applicability in LVV remains to be validated, and current evidence is still limited [103-106]“.

We also discuss new evidence in the Multimodal Assessment Section (Section 7) : “According to the EULAR 2023 recommendations, ultrasound is the first-line imaging test for suspected GCA [18]. When the result is negative or inconclusive but clinical suspicion remains high, additional imaging and/or histology is advised. The choice of the next investigation should be guided by whether cranial or extracranial involvement is suspected, local availability, and pre-test probability. Head-to-head data support this phenotype-guided, sequential strategy. In the prospective comparison by van Nieuwland et al. [139], all patients with suspected GCA underwent ultrasound, whole-body FDG-PET/CT, and cranial MRI within five days. Sensitivity was 69.6% for ultrasound, 52.2% for FDG-PET/CT, and 56.5% for MRI, with 100% specificity for all modalities. Importantly, FDG-PET/CT was negative in all isolated cranial GCA cases, while cranial MRI was negative in all isolated extracranial GCA cases. In several patients with negative or inconclusive ultrasound, subsequent imaging confirmed the diagnosis, reinforcing that ultrasound can serve as the first test and be complemented by a phenotype-guided second modality when diagnostic uncertainty persists. This principle has been operationalised in a large real-world ultrasonography-led multimodal pathway reported by Mukhtyar et al. in 1,000 consecutive referrals  [140]. The initial step consisted of scanning the superficial temporal and axillary arteries, where ultrasound positivity required involvement of ≥ two arteries (unilateral temporal-artery disease alone was considered negative, prompting extended scanning). If initial ultrasound was negative and CRP ≥ 20 mg/L, a manifestation-based extended ultrasound of additional cranial or extracranial branches was performed. When no vasculitis was found and no alternative diagnosis was evident, a second test was obtained—temporal-artery biopsy for cranial-predominant symptoms or FDG-PET/CT for constitutional/extracranial presentations. Diagnostic performance improved stepwise: 78.7% sensitivity for temporal + axillary scanning, 86.1% after extended ultrasound, and 95.7% after including a second test, with a negative predictive value of 98.0%. For comparison, temporal-artery imaging alone achieved 62.3% sensitivity. This structured, multimodal algorithm allowed safe glucocorticoid tapering when results remained negative.

Beyond diagnosis, a multimodal strategy also informs relapse monitoring. Ultrasound has shown moderate accuracy in longitudinal cohorts, and dynamic changes in ultrasound activity scores may predict relapse. Cranial MRI can reveal vascular changes not mirrored by clinical symptoms or inflammatory markers, indicating that the modalities capture different aspects of disease biology. Selective use of additional imaging at suspected relapse—rather than routine serial scanning in stable remission—aligns with EULAR recommendations for pragmatic, phenotype-guided follow-up [18].

Hybrid imaging techniques such as PET/MRI, which integrate metabolic and structural data, may further enhance diagnostic precision. In this context, multimodal quantitative tools such as the Vasculitis Activity using MR-PET (VAMP) score have been proposed to standardise disease-activity assessment by combining MRI vessel wall enhancement with PET metabolic activity to provide a reproducible, objective index of inflammation [142,143]. Together, these advances underscore the potential of multimodal imaging to improve diagnostic confidence, precision, and reproducibility in the longitudinal management of LVV.”

Lastly, in the Future Perspectives Section (Section 8, 2nd paragraph) we report on the importance of imaging in future personalised strategies: “Although a universally accepted imaging ‘gold standard’ for defining relapse in LVV is still lacking, validated indices are emerging. In GCA, the OGUS has demonstrated high reliability, sensitivity to change, and growing prognostic value across multiple cohorts. In TAK, the composite TAIDAI index - integrating clinical, laboratory, and imaging domains - has shown excellent sensitivity and good specificity in development studies. Further external validation, threshold harmonization, and linkage to patient-centred outcomes remain priorities before protocolised imaging-guided treat-to-target approaches can be widely recommended. Incorporating imaging biomarkers into treat-to-target frameworks may ultimately enable more personalised, proactive, and outcome-focused management of LVV.”

Reviewer 2 Report

Comments and Suggestions for Authors

Sopa I et al in the manuscript entitled “The Role of Imaging in Monitoring Large Vessel Vasculitis: A Comprehensive Review” provide an overview of the role of imaging in LVV. Although of interest since imaging is a useful tool for the diagnosis, monitoring and management of LVV patients there are some issues that should be addressed:

  • Ultasonography is a valuable tool assessing both arterial wall inflammation and structural damage at diagnosis and follow-up. Moreover, U/S can be used for the evaluation of CVD risk (atheromatosis, atherosclerosis, inappropriate arterial remodeling) (Argyropoulou OD et al Curr Opin Rheumatol 2018) which is higher in LVV patients than in the general population and accounts for the increased morbidity and mortality of these patients. A relevant comment should be added,
  • Although halo sign is considered pathognomonic of vascular inflammation false positive results may confuse LVV patients’ assessment. This concern should also be discussed
  • 18F FDG PET/CT detects metabolically active macrophages, thus false negative results are possible when the predominant cell population involved into the pathogenesis is other than macrophages (e.g fibroblasts). For these cases PET/CT using other cell tracers (FAPIs) etc could be used. A comment on this concern with relevant literature should also be added.

Author Response

We would like to express our gratitude for the invitation extended to Prof. Cristina Ponte to submit a revision paper on the role of imaging in monitoring disease activity in large vessel vasculitis. We are writing to resubmit the revised version of our manuscript for consideration for publication in Biomolecules.

We appreciate the opportunity to revise our work based on the insightful feedback received from the reviewers, and we believe that our revised manuscript now addresses the concerns raised during the initial evaluation process.

We have carefully considered the comments of the reviewers and prepared a point-by-point response to address each of the reviewers’ concerns.

Reviewer 2:

  1. “Ultasonography is a valuable tool assessing both arterial wall inflammation and structural damage at diagnosis and follow-up. Moreover, U/S can be used for the evaluation of CVD risk (atheromatosis, atherosclerosis, inappropriate arterial remodeling) (Argyropoulou OD et al Curr Opin Rheumatol 2018) which is higher in LVV patients than in the general population and accounts for the increased morbidity and mortality of these patients. A relevant comment should be added.”

Reply: We thank the reviewer for this important suggestion. We have added information in the Ultrasound section (Section 2, 1st paragraph) highlighting the role of ultrasonography in cardiovascular risk assessment: “Although these sonographic signs are highly characteristic of vascular inflammation, false positives can occur, most often due to overlap with atherosclerotic changes [4]. This underscores the importance of operator expertise and standardized protocols, while also highlighting the increased cardiovascular burden in LVV patients, in whom accelerated atheromatosis and arteriosclerosis contribute to excess morbidity and mortality [5].”

  1. “Although halo sign is considered pathognomonic of vascular inflammation, false positive results may confuse LVV patients’ assessment. This concern should also be discussed.”

Reply: We thank the reviewer for this valuable comment. We agree that false-positive findings may complicate the interpretation of ultrasound results in LVV. To address this, we have revised Section 2 (first paragraph) to clarify this limitation, adding the following statement: “Although these sonographic signs are highly characteristic of vascular inflammation, false positives can occur, most often due to overlap with atherosclerotic changes [4]”. In addition, we have expanded the manuscript with a new Section 7 (Multimodal assessment), which discusses how complementary imaging modalities can aid patient assessment and help resolve potential diagnostic uncertainty arising from atypical or ambiguous halo findings: “Different imaging modalities offer distinct yet complementary strengths, and their integration can enhance the interpretation of disease activity and vascular damage (Table 1). This multimodal approach is particularly valuable when multiple explanations are possible—such as differentiating vascular remodelling, atherosclerosis, and active inflammation—or when clinical and laboratory findings are inconclusive, as often occurs in patients receiving IL-6 blockade. Imaging selection should be guided by the clinical question, vascular territory involved, and disease phenotype.”

  1. “18F FDG PET/CT detects metabolically active macrophages, thus false negative results are possible when the predominant cell population involved into the pathogenesis is other than macrophages (e.g fibroblasts). For these cases PET/CT using other cell tracers (FAPIs) etc could be used. A comment on this concern with relevant literature should also be added.”

Reply: Thank you for this important point. We have now added a brief paragraph noting that alternative tracers are emerging in Fluorine-18-fluorodeoxyglucose PET section (Section 4, subsection 4.2, 4th paragraph): “In light of these limitations, as FDG uptake primarily reflects glycolytically active, macrophage-rich inflammation and may yield false-negative results in lesions dominated by other cell types such as fibroblasts, alternative tracers are being explored. Among these, fibroblast activation protein inhibitors (FAPI) and somatostatin receptor ligands (e.g., ^68Ga-DOTATATE) have shown promising preliminary results. Nevertheless, their clinical applicability in LVV remains to be validated, and current evidence is still limited [106-109].”.

Round 2

Reviewer 2 Report

Comments and Suggestions for Authors

All comments have been addressed as suggested